# Targeting Glioblastoma-Associated Macrophages for Photodynamic Therapy Using AGuIX^®^-Design Nanoparticles

**DOI:** 10.3390/pharmaceutics15030997

**Published:** 2023-03-20

**Authors:** Lucie Lerouge, Mickaël Gries, Alicia Chateau, Joël Daouk, François Lux, Paul Rocchi, Jessica Cedervall, Anna-Karin Olsson, Olivier Tillement, Céline Frochot, Samir Acherar, Noémie Thomas, Muriel Barberi-Heyob

**Affiliations:** 1Department of Biology, Signals and Systems in Cancer and Neuroscience, CRAN, UMR7039, Université de Lorraine-French National Scientific Research Center (CNRS), 54500 Vandœuvre-lès-Nancy, France; 2Institute of Light and Matter (ILM), UMR5306, Université de Lyon-CNRS, 69100 Lyon, France; 3Department of Medical Biochemistry and Microbiology, Science for Life Laboratory, Biomedical Center, Uppsala University, 75105 Uppsala, Sweden; 4Reactions and Chemical Engineering Laboratory (LRGP), UMR7274, Université de Lorraine-CNRS, 54000 Nancy, France; 5Laboratory of Chemical Physics of Macromolecules (LCPM), UMR7375, Université de Lorraine-CNRS, 54000 Nancy, France

**Keywords:** glioblastoma, photodynamic therapy, AGuIX^®^ nanoparticles, macrophages polarization, NRP-1 targeting, inflammatory effect

## Abstract

Glioblastoma (GBM) is the most difficult brain cancer to treat, and photodynamic therapy (PDT) is emerging as a complementary approach to improve tumor eradication. Neuropilin-1 (NRP-1) protein expression plays a critical role in GBM progression and immune response. Moreover, various clinical databases highlight a relationship between NRP-1 and M2 macrophage infiltration. In order to induce a photodynamic effect, multifunctional AGuIX^®^-design nanoparticles were used in combination with a magnetic resonance imaging (MRI) contrast agent, as well as a porphyrin as the photosensitizer molecule and KDKPPR peptide ligand for targeting the NRP-1 receptor. The main objective of this study was to characterize the impact of macrophage NRP-1 protein expression on the uptake of functionalized AGuIX^®^-design nanoparticles in vitro and to describe the influence of GBM cell secretome post-PDT on the polarization of macrophages into M1 or M2 phenotypes. By using THP-1 human monocytes, successful polarization into the macrophage phenotypes was argued via specific morphological traits, discriminant nucleocytoplasmic ratio values, and different adhesion abilities based on real-time cell impedance measurements. In addition, macrophage polarization was confirmed via the transcript-level expression of TNFα, CXCL10, CD-80, CD-163, CD-206, and CCL22 markers. In relation to NRP-1 protein over-expression, we demonstrated a three-fold increase in functionalized nanoparticle uptake for the M2 macrophages compared to the M1 phenotype. The secretome of the post-PDT GBM cells led to nearly a three-fold increase in the over-expression of TNFα transcripts, confirming the polarization to the M1 phenotype. The in vivo relationship between post-PDT efficiency and the inflammatory effects points to the extensive involvement of macrophages in the tumor zone.

## 1. Introduction

Glioblastoma (GBM) is recognized as being extremely challenging to treat due to its aggressiveness and the intrusive properties of the tumor cells in the adjacent brain parenchyma. As a result, the prognosis is poor due to inevitable recurrences, making it the most aggressive primary brain tumor in adults. Standard treatment consists of surgical resection followed by postoperative radiotherapy combined with temozolomide chemotherapy [1]. The wide majority of GBMs recur, and patients often succumb to and die from local recurrences, indicating that more aggressive local therapy is required for better eradication. In this unfavorable context, photodynamic therapy (PDT) appears to be a complementary approach that may reduce recurrence and extend survival with minimal side effects [2].

PDT aims to destroy neoplastic lesions by the combined action of a photosensitizer (PS) and visible light, leading to the formation of reactive oxygen species (ROS), especially singlet oxygen. Several clinical trials were reported, suggesting different approaches, such as interstitial PDT (iPDT) and intraoperative PDT. Using iPDT guided by real-time imaging, we demonstrated the major role of the antivascular effect in GBM eradication [3]. We already highlighted the potential of ultrasmall AGuIX nanoparticles named AGuIX@PS@KDKPPR, consisting of a surface-localized ligand peptide (KDKPPR) targeting neuropilin-1 (NRP-1), porphyrin as the PS, and gadolinium as the magnetic resonance imaging (MRI) contrast agent [4,5]. In order to attest KDKPPR-conjugated nanoparticle affinity, we performed surface plasmon resonance (SPR) assays, validating a dissociation constant (K_D_) value of 4.7 µM for human NRP-1 protein [5]. This transmembrane receptor was suggested as a molecular therapeutic target for GBM [6] as its overexpression is mainly due to angiogenic endothelial cells and glioma-associated macrophages (GAM), with both being related to an unfavorable prognosis [7]. In vitro studies revealed that tumor necrosis factor alpha (TNFα) is a relevant growth factor involved in PDT efficiency [8,9]. When taken together, these results also suggest that the immunostimulatory effects triggered by PDT may modulate macrophage phenotype toward a proinflammatory and immunogenic M1 polarization.

In this context, GAMs are subverted by tumor cells and are widely responsible for orchestrating tumor progression through the secretion of factors promoting chemoattraction, immune suppression, angiogenesis and tumor cell survival [10,11]. GAMs are also recognized as an integral part in GBM progression [12,13]. It was reported that either population of microglia or peripheral macrophages, lacking NRP-1, were able to inhibit GBM progression [14,15]. In this context, a comprehensive understanding of the mechanism of PDT-induced macrophages activation and its role in GBM response to radiation therapies remains crucial. Moreover, nanoparticles can also disrupt polarization and macrophage reprogramming, affecting their immunological function and modifying post-treatment response [16,17,18].

GBM has a high inflammatory potential, with macrophages (M0) being the most important reactive cell population. Two phenotypes can be obtained with the polarization of M0: the M1 anti-tumoral population and the M2 protumoral population. Various external factors may influence polarization, meaning the three phenotypes (M0, M1, and M2) can cohabit in the tumor microenvironment. Macrophages can reach 30% of the total cells in low-grade gliomas [19] and in GBM [10]. By releasing chemoattractant factors, including monocyte chemoattractant protein-1 (CCL2/MCP-1) or even stromal cell-derived factor 1 (CXCL12/SDF-1), tumor cells actively recruit circulating monocytes and facilitate their migration to the tumor area [20]. When activated by, for instance, interferon-gamma (IFNγ), TNFα and lipoproteins, M1 phenotype macrophages perform mainly antitumor functions due to their phagocytic property, ROS production, and their ability to activate the synthesis of proinflammatory cytokine interleukins (ILs), such as IL-1β, IL-12, and TNFα [21]. As actors of adaptive immunity, M1 macrophages also develop the capacity to present antigens to T cells [22]. In contrast, the IL-4, IL-10, IL-13, and transforming growth factor beta (TGF-β) present in the tumor microenvironment polarize macrophages into the M2 phenotype. These are well-known to promote tumor development through tissue remodeling; they secrete proteases, such as matrix metalloproteinase-2 (MMP-2) and matrix metalloproteinase-9 (MMP-9) [23]. Moreover, M2 macrophages secrete vascular endothelial growth factor (VEGF) and potent immunosuppressive cytokines, including IL-10, IL-6, and TGF-β, leading to an enabling environment for tumor proliferation [24]. Although the different phenotypes are found in the GBM stroma, the M2 phenotype is predominant with a preferential localization in hypoxic and anoxic areas, while the M0 and M1 phenotypes are mainly found in normoxic areas [25]. An M1/M2 ratio in favor of the M2 phenotype was correlated with aggressiveness and poor prognosis in patients with GBM [21]. Interestingly in nonmalignant brain tumors, most of the macrophage population and microglial cells were found to be M1, increasing proinflammatory activity and promoting tumor lysis [20].

In this article, using a U87 cell xenografted *nude* rat model, we confirmed that M2 was the major macrophage phenotype in GBM tumors stroma, arguing the interest of an NRP-1 targeting strategy using AGuIX@PS@KDKPPR nanoparticles. By taking THP-1 human monocyte cells as a model for macrophage polarization [26], we documented their successful differentiation between the M1 and M2 phenotypes by complementary approaches, combining morphological analysis and adhesion characteristics, as well as transcripts expression. Following this successful discrimination between M1 and M2 macrophages, we demonstrated that high NRP-1 protein overexpression in M2 macrophages led to a statistically significant uptake of functionalized nanoparticles with KDKPPR ligand peptides. Furthermore, to assess the PDT immunostimulant effect, M0 were incubated with U87 cell secretome post-treatment, leading to M1 polarization. Via in vivo MRI, we highlighted a post-PDT inflammation effect with macrophage recruitment, supporting a promising irradiation schedule.

## 2. Materials and Methods

### 2.1. Cell Culture

#### 2.1.1. THP-1 Cell Polarization Process

Nonadherent THP-1 cells (ATCC^®^ TIB-202™) were grown in an RPMI 1640 medium (Roswell Park Memorial Institute) supplemented by 10% *v*/*v* of fetal calf serum (FCS) uncompleted (Sigma-Aldrich, St Quentin Fallavier, France), 1% *v*/*v* antibiotics (penicillin at 100 U·mL^−1^ et streptomycin at 100 µg·mL^−1^, Sigma-Aldrich) and 2-mercaptoethanol for a final concentration of 0.05 nM (Sigma-Aldrich).

THP-1 monocytes were differentiated into adherent macrophages (M0) by adding 320 nM phorbol-12-myristat-13-acetate (PMA) (Sigma-Aldrich). At 24 h post-PMA incubation, the culture medium was changed by adding 100 ng·mL^−1^ lipopolysaccharide (LPS, R&D systems, Minneapolis, MN, USA) and 20 ng·mL^−1^ interferon gamma human recombinant (IFN-γ, R&D systems) for an M1 polarization. For M2, 20 ng·mL^−1^ interleukine-4 human recombinant (IL-4, R&D systems) and 20 ng·mL^−1^ interleukine-13 human recombinant (IL-13, R&D systems) were added. After 72 h, the cells were used for experiments.

#### 2.1.2. U87 Tumor Cell Line

U87-MG cells (ATCC^®^ HTB-14™) were grown in a DMEM medium (Dulbecco Modified Eagle Medium) with 4.5 g·L^−1^ glucose (Gibco, France), supplemented by 10% *v*/*v* of FCS uncompleted, 5% supplement containing nonessential and essential amino acids, vitamins and sodium pyruvate, and 1% *v*/*v* antibiotics (Penicillin at 100 U·mL^−1^, Streptomycin at 100 µg·mL^−1^) (Gibco).

### 2.2. Macrophages Characterization Pre- and Post-PDT

#### 2.2.1. PDT Protocol

To investigate the effect of the secretary factors from the U87 cells on macrophages, a 2D coculture model was performed. Medium secreted by the U87 with or without PDT (secretome) was recovered and brought into contact with macrophages to analyze the potential effects on their polarization. Trypsinization on U87-MG cells (grown as explained in “Cell culture”) was performed to seed 6 × 10^4^ cells·mL^−1^ in T75 flasks in DMEM medium. Four days later, U87-MG cells were grown at 4 × 10^4^ cells·mL^−1^ in P6 plates (Falcon^®^ Cell culture 6-well plate, Thermo Fisher Scientific), with 2 mL per insert. In parallel, THP-1 cells were seeded the same day as U87-MG, at 8 × 10^5^ cells·mL^−1^ (grown as explained in “Cell culture”) in T150 flasks.

Three days later, from those same THP-1 cells, M0 cells were grown at 9 × 10^5^ cells·mL^−1^ in P6 plates. U87 cells were treated with NPs at 1 µM over 4 h, which is the drug-light interval (DLI), followed by a medium change and PDT at 10 J·cm^−2^ (652 nm, 700 mW fiber output power, 4.54 mW·cm^−2^ irradiance) for 2202 s. It was performed using a laser diode (Laser Ceralas™ PDT, Biolitec, Jena, Germany). After 24 h, the secretome was recovered by centrifugation for 10 min at 300× *g*, which was then added (80%) to the M0 5X medium (20%) in the P6 wells containing M0. In parallel, a negative control corresponding to untreated U87 cell secretome was added onto M0 as well. At 24 h post-contact with U87 secretome, an RNA extraction was performed on all P6 wells, and microscopy analyses were used for cytoplasmic labeling (CellVue, Sigma-Aldrich) and nucleocytoplasmic ratios calculation by confocal microscopy.

#### 2.2.2. Cell Morphology Analysis

After their polarization and fixation with paraformaldehyde 4% (Thermo Fisher Scientific, Illkirch, France), macrophages were labelled with hematoxylin (Sigma-Aldrich). Cells were observed with the Nikon Eclipse E600 optical microscope 7 days after polarization for optimal morphologic features.

#### 2.2.3. Real-Time Cell Impedance Measurements

The proliferation, morphology (shape, size), and adhesion of the cells as a function of the differentiation/polarization media or conditioned media were evaluated on the XCELLigence (Agilent Technologies, San Diego, CA, USA). For the preparation of the conditioned media, U87 cells were seeded at 2000 cells·mm^−2^ in a T25 flask. When the cells had reached 90% confluence (3–4 days), they were treated (see Section 2.2.1). In parallel, a negative control corresponding to untreated cells was performed. At 24 h post-PDT, the culture medium was removed and centrifuged at 300× *g* for 10 min. Since the U87 and THP-1 cells were grown in 2 different cultured media (DMEM for U87 and RPMI 1640 for THP-1), a mixture was, thus, made, containing 80% of the secretome of U87 cells treated (or not) with PDT and 20% of M0 medium concentrated five times. This mixture was added to the M0 cells following impedance measurements for 72 h. In parallel, impedance measurements were performed on the M0, M1, and M2 cells. To do so, THP-1 cells were seeded at 3000 cells·mm^−2^ in P96 plates (Agilent Technologies) in M0 medium for 24 h at 37 °C and were differentiated into M1 and M2 (see Section 2.1.1).

#### 2.2.4. Nucleocytoplasmic Ratio

To calculate the nucleocytoplasmic ratio for each macrophage phenotype and to quantify the cell coverage area, cells cytoplasm was labeled using a lipidic marker called CellVue^®^ Claret Far Red Fluorescent Cell Linker Kits (Sigma-Aldrich). Visualization was performed with a fluorescent microscope (ImageXpress^®^, Molecular Devices, San Jose, CA, USA). Cells were incubated for 3 min at 37 °C before the labeling solution was removed. Cells were washed two times with M0 medium. M0, M1, M2, U87 secretome pre- (control) and post-PDT (PDT 10J) medium, related to culture conditions, was added to the corresponding well. ImageJ software was used to analyze the resulting images using the ROI tool (region of interest) to select cells one by one; the cytoplasm (Cy5) and nucleus (DAPI) images were separated, and a threshold was applied to each one to detect the outline of the nuclei and cytoplasm. Then, using a measurement tool, the values were obtained, and the ratio was calculated.

#### 2.2.5. Phenotype Characterization by Gene Markers

RT-qPCR experiments were used to characterize each macrophage phenotype using specific primers to bind to the chosen targets. First, cells were washed with PBS buffer after removing the culture medium, then GTC Lysis Buffer (OMEGA Bio-Tek, Norcross, GA, USA) was added; cells were then scraped and collected (stored at −80 °C if not used directly). EZNA^®^ HP Total RNA Kit (OMEGA Bio-Tek) allowed us to isolate the total RNA from the cultured cells with effective genomic DNA elimination by using mini spin columns. An “RNA Homogenizer Mini Column” and an “HiBind RNA Mini Column” were used (detailed protocol on OMEGA bio-Tek). After the extraction, ultra-pure water (Nuclease Free Water, OMEGA bio-tek) was added to the eluate samples. RNA dosage was performed with the NanoDrop One spectrophotometer (Thermo Fisher Scientific). 

RT-qPCR is carried out on samples at 25 ng·µL^−1^ with the iScript RT SupermixTM (Bio-Rad, Hercules, CA, USA) containing the reverse transcriptase (RT) enzymes, primers, and dNTPs. Priming was carried out at 95 °C for 5 min, reverse transcription at 46 °C for 20 min, followed by an RT inactivation at 95 °C. Each sample was diluted ten times (2.5 ng·µL^−1^ of cDNA) to achieve the qPCR step with the iTaq SYBR^®^ Green kit (BioRad, Hercules, CA, USA, at 2X) containing hot start iTaq DNA polymerase, dNTPs, SYBR^®^ Green dye, and MgCl_2_. Forward and reverse primers for the gene of interest (Table 1) were added at 1/10 to the mix with milli-Q water (qsp 20 µL). Polymerase activation was performed at 95 °C for 1 min, denaturation of samples at 95 °C for 5 sec, followed by the annealing of the primers at the hybridization temperature (60 °C for TNFα, CD-163, CCL22, GAPDH (glyceraldehyde-3-phosphate dehydrogenase), CD-80, CD-206 and 57 °C for CXCL10) and elongation/amplification during 40 cycles by heat-resistant Taq polymerase. The analysis method used is the quantity-based method, where RNA quantity was obtain from the Cq values, thanks to the standard range, and are reported as GAPDH quantity (housekeeping gene). For each gene, the first quantity value was reported as 1, and the average was calculated (for duplicates and then for *n* = 3 experiments).

At the end of the PCR, 10 μL of the amplification products was added to 2 μL of deposition blue, and this was plated on a 2% agarose gel to verify the presence of a single amplicon of the expected size. To check for the absence of RNA degradation and genomic DNA contaminants, 500 ng of these extracts was added to 2 μL of blue deposition, which was spotted onto a 1.2% agarose gel with a fluorescent intercalant for nucleic acid detection (Gel Red Nucleic Acid Gel Stain, Interchim, Monluçon, France). Migration was performed at 140 volts for 30 min in TAE 1X buffer (Invitrogen, Paisley, UK).

### 2.3. Secretome Analysis Pre- and Post-PDT

To identify the cytokines released in the secretome of U87 cells (that could possibly contribute to a preferential polarization into M1 or M2 macrophages), we used a proteome array analysis. This approach was also performed on the post-PDT U87 cells.

As in Section 2.2.1, the culture medium was recovered 24 h post-PDT and centrifuged at 300× *g* for 10 min. It was analyzed using the Human Cytokine Antibody Array kit (Abcam, Cambridge, UK) (Table 2). Membranes were blocked with 2 mL of 1X Blocking Buffer at room temperature for 30 min before depositing the samples on membranes. To increase signals, it is recommended that samples be incubated overnight at 4 °C. Membranes were then washed with 20–30 mL of Wash Buffer I for 45 min, followed by 3 washes with 2 mL of Wash Buffer I (1X) and 3 additional washes with Wash Buffer II (1X) for 5 min each. Biotin-conjugated 1X anti-cytokine mixture was added, followed by HRP-conjugated 1X streptavidin mixture overnight at 4 °C. Chemiluminescence revelation was performed using the LAS-4000 software (Fujifilm, Tokyo, Japan). Data were quantified using the Multi-Gauge software. It was used to process the resulting images by drawing an ROI (region of interest) on each spot. The background (negative spots) was deducted, and in order to compare each different membranes, the results were normalized by the average intensity of the positive control.

### 2.4. Macrophages Targeting by AGuIX@PS@KDKPPR

#### 2.4.1. Nanoparticles Uptake

Uptake of functionalized nanoparticles was assessed to determine the impact of NRP-1 targeting between the macrophage phenotypes. THP-1 cells were inoculated in T150 flasks (Dominique Dutscher, Bernolsheim, France) at 2 × 10^5^ cells·mL^−1^, differentiated, and polarized 24 h later (seen in Section 2.1.1). At 72 h post-inoculation, cells were brought into contact with AGuIX@PS@KDKPPR or AGuIX@PS@scramble for 4 or 24 h in the dark at 1 µM and at 37 °C. After incubation, the cells were washed with HBSS buffer (Sigma-Aldrich) and were detached by scraping. Cell suspensions were centrifuged for 7 min at 120× *g*, with the recovery of the pellet in 500 µL of PBS, and 40 µL of the volume saved for automatic counting with the TC20™ (Automated Cell Counter, Bio-Rad). Cell lysis was performed by sonication for 15 min in the presence of a lyse buffer composed of PBS, 0.5% *v*/*v* Triton 100X (Prolabo, Bern, Switzerland), and 8 mg·mL^−1^ sodium hydroxide (Merck, Darmstadt, Germany). After centrifugation for 15 min at 10,000× *g*, a fluorescence reading (excitation: 415 nm, emission: 650 nm) was taken of the supernatant (TECAN Infinite^®^ M200 PRO, Mannëdorf, Switzerland). A calibration curve of the porphyrin grafted onto the nanoparticles was performed in the lysis buffer at concentrations ranging from 1 to 100 nM. Each condition was performed five times in the dark.

#### 2.4.2. NRP-1 Protein Detection

To further validate the macrophage markers, an immuno-cytofluorescence method was used using fluorescent antibodies to identify the potential markers. Cell fixation was carried out using PAF 4% at 20 °C for 30 min, followed by two 5 min-PBS washes. Saturation step was carried out with PBS with BSA 3% for 1 h, followed by the deposition of NRP-1 antibodies (Abcam) diluted at 1:250 in PBS with BSA 1% overnight at 4 °C. Secondary antibody was diluted at 1:1000 (anti-rabbit, Thermo Fisher Scientific) in PBS with BSA 1% and incubated for 1 h. Hoechst labeling was carried out at 1:1000 in PBS for 15 min. Each step was separated by 1 or 2 washes for 5 min in PBS.

#### 2.4.3. Macrophages Survival Pre- and Post-PDT

THP-1 cells were seeded at 10^5^ cells·mL^−1^ in P96 plates, and, as described in Section 2.1.1, cells were differentiated and polarized into the M0, M1, or M2 phenotypes. Different treatments were performed. Nanoparticles were added at a concentration of 1 µM for 4 h or 24 h, and the PDT was carried out at 10 J·cm^−2^ (same as Section 2.2.1). Two control groups were performed, “NPs alone”, with only the addition of 1 µM of AGuIX@PS@KDKPPR, and “Light alone”, with only laser exposition (652 nm, 700 mW fiber output power, 4.54 mW·cm^−2^ irradiance). At 24 h post-treatment, an MTT solution (Thermo Fisher Scientific) was added at 2.5 mg·mL^−1^ to each well. At 3 h post-incubation at 37 °C and 5% CO_2_ in the dark, the medium was removed, and DMSO (Thermo Fisher Scientific) was added to allow the dissolution of the formed formazan crystals. The reading was performed using a spectrophotometer at 540 nm (Multiskan Ascent Thermo, Madison, WI, USA). Each condition was performed in quadruplicate, and the results were given as mean ± standard deviation (SD).

### 2.5. In Vivo Experiments

The use of experimental animals during the in vivo experiments was carried out with respect to the 3 R requirements for animal welfare, and these actions were carried out by competent and authorized persons in a registered establishment (numbers D-54-547-03 issued by the Departments of Veterinary Services). It was performed in accordance with the European Community animal care guidelines (Directive 2010/63/EU), with all protocols and procedures examined and validated by an internal ethics committee (Comité d’Ethique Lorrain en Matière d’Expérimentation Animale, CELMEA, French Ethical Committee number 66). According to national regulations, the research project APAFIS #20985 was approved by the French Ministry of Research. During the hosting and experiments, the animals were kept under standard conditions, which were 24 °C ± 1 °C, 50% ± 10% hygrometry, and a controlled 12 h light–dark cycle. For the model, we used six-week-old nude male rats (Hsd: RH-Foxn1rnu) who were purchased from Envigo (Horst, The Netherlands). The orthotopic xenograft model (in the right caudal putamen of rats) has been previously described in [3] for the graft and cranial anchor model.

#### 2.5.1. Immuno-Histochemical Analysis on Tissues and Frozen Sections

Formalin-fixed tumor tissue was embedded in paraffin and cut into 5 μM sections. Subsequently, these sections were stained with H&E (Hematoxylin & Eosin) following standard procedures. Tissue sections were deparaffinized in toluene and rehydrated with graded alcohols. Antigen recovery was performed by heating slides in sodium citrate buffer (pH 6.0). The labeling of CD-68, CD-86, CD-163, and CD-206 (Thermo Fisher Scientific) was performed with a dilution of 1:1500, 1:5000, 1:500, and 1:3000, respectively, at 4 °C overnight. The slides were treated with 1:5 hydrogen peroxide (H_2_O_2_) for 20 min to quench the endogenous peroxidase activity. The sections were washed with PBST and incubated with the secondary antibody Histofine^®^ (Nichirei Biosciences, Tokyo, Japan) for 20 min. Finally, labeling was revealed by Novared^®^ (Vector Laboratories) with a conventional hematoxylin counterstain. The number of positive cells for each marker was automatically quantified using QuPath software (v.02.3). Three sections were performed on the same tumor with three different images for each section.

Brain tissues were also recovered 1 h, 24 h, and 48 h after PDT and were frozen in liquid-nitrogen-cooled isopentane. Briefly, the organs were embedded in the OCT (Optimal Cutting Temperature compound) and 8 µm-thick cuts were made on a cryostat. The sections were dried and fixed in ice-cold acetone for 10 min. The slides were stained with antibodies to CD68 (Invitrogen, 11-0689-42) in combination with a fluorophore FITC (fluorescein isothiocyanate). Counterstaining was performed. Hoechst counterstaining was performed to visualize the cell nucleus.

#### 2.5.2. PDT-Guided by MRI

AGuIX@PS@KDKPPR nanoparticles were injected intravenously into the tail vein (1.75 µmol·kg^−1^, porphyrin equivalent) in U87 tumor-bearing *nude* rats. To allow for the illumination of the tumor, an optical fiber (UltraSil 272 ULS, OFS) was inserted into the brain, as previously performed in Gries et al. [5]. Illumination was performed at 652 nm for 8 min 40 s (20.8 J) with a fiber output power of 40 mW. Different groups were followed: (1) rats exposed to light after injection of AGuIX@PS@KDKPPR (*n* = 3), (2) rats injected only with AGuIX@PS@KDKPPR (*n* = 5), and (3) rats without any treatment (*n* = 6).

#### 2.5.3. MRI Images Acquisition

A longitudinal MRI study was performed using a 7 Tesla small animal MRI (Biospec 70/20 USR, Bruker, Wissembourg, France). A transmitting body antenna and a receiving head antenna were used for all acquisitions. Paravision 5.2 software (Bruker) was used to analyze the data. The parameters of the MRI sequences for the T2-weighted imaging were TR/TE: 5000/77 ms, FOV: 4 × 4 cm, matrix: 256 × 256, and pixel size: 0.16 × 0.16 mm. For Diffusion Weighted Imaging (DWI), the parameters were TR/TE: 3000/30 ms, FOV: 4 × 4 cm, matrix: 128 × 128, pixel size: 0.23 × 0.27 mm, and b factor: 100, 200, 400, 600, 800, and 1000 s·mm^−2^. DWI sequences allowed for us to map the apparent diffusion coefficients (ADCs), which are related to the random motion of water molecules according to tissue cellularity and the presence of an intact cell membrane.

### 2.6. Statistical Analysis

Statistical analyses were conducted through Kruskal–Wallis test and Mann–Whitney tests (for Figure 6c only) using GraphPad Prism 5 (GraphPad Software, San Diego, CA, USA). All results were mean ± standard deviation (SD).

## 3. Results

### 3.1. In Vivo Recruitment of M2 Macrophages in the Tumor Stroma of GBM

In order to verify whether the resident macrophages in the GBM tumor environment were indeed protumor, as is widely described in the literature, we performed immune-histochemical labeling on the tumor tissue of the U87 xenograft rat models sections to assess the percentage of M2 macrophages in the overall population (Figure 1a–d). CD-163 and CD-206 are M2 markers [27], and CD-86 is mainly considered a marker for a subtype called M2b [28,29]. We determined the mean values at about 15%, 12%, and 19% using CD-86, CD-163, and CD-206 labeling, respectively (Figure 1e). When considering the percentage of positive cells (CD-68 labeling; marking macrophage membranes), only the M2 macrophages were almost exclusively recruited, representing a mean of 75% for the macrophagic population. CD-68 is the only known member of the class D scavenger receptors and is a valuable cytochemical marker for immunostaining monocytes and all macrophages.

### 3.2. Successful Polarization of THP-1 in M1/M2 Phenotypes by Morphological Features and Adhesion Capacities

#### 3.2.1. Macrophage Morphology

In order to characterize the macrophage phenotypes, optical microscopic observations were performed after the polarization process. The M0 cells were characterized by roundish cell morphology, lacking cytoplasmic extensions, whereas the M1 phenotype macrophages highlighted extensive cytoplasmic extensions (Figure 2a). The macrophages with an amoeboid morphology were characterized as the M0 and M2 phenotypes in contrast to the M1 macrophages, with a typical dendritic morphology (Figure 2a).

Real-time cell impedance measurements were used to characterize the impact of macrophage polarization on the adhesion characteristics. As expected, during the differentiation phase, no difference was evidenced (Figure 2b). The second step was a 48 h incubation in a polarization media containing LPS and IFNγ for the M1 phenotype or IL-4 and IL-13 for M2. The M0 macrophages were maintained in the differentiation medium. During the polarization phase, the cell index values of the different macrophage populations rapidly evolved in very different ways: for the M1 macrophages, they remained stable, and the area under the curve mean values were reduced compared to M0, with 2.49 ± 0.45 versus 6.46 ± 1.42, respectively (Figure 2b). On the contrary, the cell index values for M2 increased with the area under the curve mean values, with 9.49 ± 1.77.

With respect to cytoplasmic expansion in the M2 macrophages and the different cell profiles identified by the impedance analysis, we calculated the nucleocytoplasmic ratios inherent to the three phenotypes (Figure 3a). As expected, we highlighted (from morphological observations) statistically significant lower cytoplasmic ratio values for the M2 phenotype macrophages (mean value of 0.27) compared to M0 (0.42) and M1 (0.37) (Figure 3b). The impedance-based detection technology measured the net adhesion of the cells to the high-density gold electrode arrays. The strength of cellular adhesion was influenced by cell phenotype and spreading. As the M2 macrophages displayed a rounded shape compared to M1 (Figure 2a), the contact surface increased.

#### 3.2.2. Macrophage Gene Expression

In order to complete the macrophage phenotype characterization, RT-qPCR experiments were conducted to evaluate the transcript expression of the relevant markers, such as TNFα, CXCL10, CD-80, CCL22, CD-163, and CD-206. Gene expression for TNFα was about eight times higher when expressed in the M1 phenotype macrophages compared to M0 and M2. The CD-80 and CXCL10 markers were over-expressed in the M1 phenotype (Figure 4). For the CD-163 gene, the transcript expression in the M2/M0 macrophages was approximately four times higher compared to M1, and the CD-206 marker was six times more expressed for the M2 phenotype. Regarding CCL22, it was only expressed in the M2 phenotype (Figure 4).

### 3.3. Targeting M2 Macrophages with AGuIX@PS@KDKPPR Nanoparticles

#### 3.3.1. Nanoparticle Uptake by Macrophages

First, we assessed (by fluorescence microscopy) the phagocytic capacity of the M1 and M2 populations exposed to the AGuIX@PS@KDKPPR nanoparticles. After 4 h of incubation, no statistical difference in nanoparticle uptake was evidenced; on the contrary, after 24 h of exposure, the KDKPPR-functionalized nanoparticles were incorporated three times more by M0 and M2 than the M1 macrophages (Figure 5a). Moreover, using functionalized AGuIX-nanoparticles with a scrambled peptide (permutation of original KDKPPR peptide sequence) as a negative control allowed us to check the uptake of KDKPPR- nanoparticles mediated by the NRP-1 receptor (Figure 5b).

#### 3.3.2. NRP-1 Protein Expression Regarding Macrophages Phenotype

In the second step and via immunofluorescence analysis, we evaluated NRP-1 protein expression in the M0, M1, and M2 phenotypes (Figure 5c,d). The quantification of the fluorescence intensity values was performed using MetaXpress software, allowing for measurements of the average labeling intensities of each cell (nuclei labeled with Hoechst). According to the mean fluorescence intensity values, the M2 phenotype macrophages expressed three times more NRP-1 protein compared to M1 (Figure 5d), which was in accordance with the 3-fold increase in nanoparticle uptake in M2.

#### 3.3.3. Effect of PDT on Macrophages

We investigated PDT efficiency on the different macrophage phenotypes using AGuIX@PS@KDKPPR. We measured no difference in cell survival between the phenotypes for PDT performed after 4 h of nanoparticle exposure (Figure 6a). On the contrary, there were statistically significant differences for the M0 and M2 macrophages compared to light alone (Figure 6b). These results (Figure 6c) are in accordance with the NRP-1 higher gene expression for the M0 and M2 phenotypes (Figure 5d) and AGuIX@PS@KDKPPR uptake (Figure 5b).

### 3.4. Post-PDT Secretome from U87 on Macrophages

Real-time cell impedance measurements were used to characterize the impact of the secretome obtained from U87 post-PDT on macrophage polarization. Macrophages were exposed to U87 cell secretome before and after PDT. The contact of M0 macrophages with untreated U87 cell secretome slightly altered their cell index evolution to 14.93 ± 3.00 (Figure 7a). On the contrary, cell index evolution highlighted that the tumor U87 cell secretome obtained post-PDT contributed to polarizing the macrophages to the M1 phenotype. (Figure 7a). These results suggested that PDT altered the secretome from the U87 GBM cells, shifting macrophage morphology toward an M1 phenotype. This analysis provided general information on the morphological variations of the macrophages. This was completed by nucleocytoplasmic ratio analysis, showing that the macrophages subjected to U87 secretome obtained post-PDT were closer to an M1 nucleocytoplasmic ratio (values at about 0.35; M1 mean nucleocytoplasmic ratio value being 0.37) (Figure 7b).

In order to confirm that the post-PDT U87 secretome polarized the macrophages toward an M1 phenotype (as shown above, Section 3.4), the RNA was extracted from macrophages, and RT-qPCR analysis was performed (Figure 8a). The TNFα gene expression level was similar to the M1 phenotype after PDT. However, it was not confirmed for CXCL10 gene expression, with just a slight increase in expression after PDT. For the M2 markers, the secretome recovered post-PDT did not seem to favor the M2 phenotype, with the absence of CCL22 marker expression and a decrease in CD-163 marker post-PDT compared to the control. In order to explain these results, we decided to analyze cytokine expression (CCL2, CXCL2, VEGF, IL-6, and IL-8) from U87 secretome pre- and post-PDT. No significant statistical differences were noticed (Figure 8b).

### 3.5. In Vivo Macrophages Infiltration Post-PDT

According to the in vitro results, PDT using AGuIX@PS@KDKPPR led to M1 polarization (Figure 7 and Figure 8) and M2 targeting (Figure 5 and Figure 6). In order to characterize the in vivo tumor response post-PDT on U87 xenografted *nude* rats and to understand how NPR-1 targeting could improve treatment efficiency, we performed MRI analysis and DWI (diffusion-weighted imaging) acquisitions.

After the intravenous injection of the AGuIX@PS@KDKPPR nanoparticles, PDT induced an inflammatory response, visualized by the apparent diffusion coefficient (ADC) maps, demonstrating peritumoral cytotoxic edema surrounded by vasogenic edemas in the tumor tissue (Figure 9a). After treatment, an inflammatory response was observed in all animals (Appendix A); The ADC values progressively increased at the tumor site, indicating an opening of the blood-brain barrier (BBB). Cytotoxic edema at the periphery of the tumor mass characterized the intracellular accumulation of water molecules with reduced diffusivity. In addition, we evidenced a major increase in macrophage recruitment post-PDT (Figure 9b).

## 4. Discussion

We recently designed AGuIX-like nanoparticles covalently grafted with a KDKPPR ligand peptide that targets the VEGF receptor NRP-1 [4]. NRP-1 is viewed as a compelling therapeutic target in GBM and, thus, has received significant attention in past years. GBM biopsies have revealed that the increased expression of NRP-1 is associated with malignancy, whereas reducing its expression suppresses migration, proliferation, and survival in vitro and stem cell viability and tumor growth in vivo [15,30]. This receptor, which is over-expressed by the angiogenic endothelial cells of tumor vasculature, is also mainly implicated in GAM immune polarization, exhibiting an increase in antitumorigenic GAM infiltrate. Investigating if KDKPPR-functionalized AGuIX nanoparticles can target macrophages and how PDT can promote their recruitment is crucial for characterizing inflammatory responses post-treatment.

In some highly inflammatory tumors, such as GBM, M0 macrophages represent the most abundant population of reactive cells. In this study, we demonstrated that M2 phenotypes were predominant in the tumor mass, as various external factors influence M0 polarization. We used CD-163 and CD-206, which are well-known to be relevant M2 markers, and CD-86, which is both an M1 and M2b marker, which could explain its higher expression level compared to the one found for CD-163 [31,32]. We investigated the in vitro analysis of the direct response of THP-1 monocyte cells. In order to highlight their polarization, we analyzed their morphology, adhesion capacities, nucleocytoplasmic ratio, and transcript marker expressions. After the polarization process, characterization using light microscopy allowed us to discriminate and differentiate the different macrophages; those incubated with IFNγ and LPS acquired a dendritic morphology compared to the M0 and M2 phenotypes, promoting rhomboid histology. These morphological and cytological features have been described previously by other teams [33,34]. Indeed, Bertani et al. labeled M1 and M2 macrophages with phalloidin conjugates to observe actin filaments, and they noticed that the M2 phenotype revealed an extended cytoplasm, and the M1 macrophages some cytoplasmic “arms”. Rostam et al. labeled actin filaments and deduced the same conclusions. We confirmed a cytoplasmic spreading for M2 in relation to nucleocytoplasmic ratio values. Moreover, these results were corroborated by an original impedance analysis, demonstrating, for the M2 phenotype, an improvement in their adhesion ability compared to M1 but not their proliferation [18]. The progressive reduction in the proliferative activity of THP-1 cells was correlated with an increase in adhesion properties.

In order to further discriminate the M1 and M2 macrophages, an analysis of the mRNA expression levels of M1 (TNFα, CXCL10, and CD-80) and M2 (CD-163, CCL22, and CD-206) markers was performed. The same markers were suggested in another study that used immunochemistry and immunofluorescence experiments [35]. In addition, regarding the gene expression profiles, M1 polarization was also evidenced for macrophages exposed to the secretomes from the U87 cells treated by PDT. Various studies involving the use of secretome from PDT-treated tumor cells on macrophages have been described [36,37]. For example, the addition of supernatant from PDT-treated EMT-6 cells (a murine breast carcinoma cell model) to RAW 264.7; macrophages increase nitric oxide (NO) production, characterizing an inflammatory macrophage activation [37]. Another team highlighted the modulation of the expression of antigenic complexes (MHCI and MHCII) and the costimulatory molecules (CD40, CD80, and CD86), involved in T cell activation [36]. These different M1 markers were widely increased by macrophages (J774A.1) after incubation with secretome from PDT-treated cells (EMT6) compared to nonactivated cells.

We used proteome array technology for secretome analysis, such as VEGF, CXCL2, CCL2, IL-6, and IL-8. However, a lack of difference between the secreted factors from the U87 tumor cells (with or without PDT) could argue for the potential impact of ROS following PDT, as well as DAMP (damage-associated molecular pattern) production. HSP70 and ATP (both DAMP) were documented as being released by dye-treated cells and providing immunostimulant effects that could explain M1 phenotype polarization. It has been published that HPS70 was expressed by PDT-treated cells and triggered the Toll-like receptor 2 and 4 (TLR2 and TLR4) signaling pathways, leading to NFκB activation [38]. The activation of NFκB signaling upregulated the generation of NO and TNFα by macrophages in addition to increasing the synthesis of other cytokines and inflammatory mediators, suggesting M1 phenotype acquisition.

In this study, we especially observed that macrophage phenotypes mainly influenced the uptake capacity of our targeting nanoparticles. Indeed, AGuIX@PS@KDKPPR were preferentially incorporated by M2 macrophages compared to the M1 phenotype. This difference in uptake capacity was consistent with NRP-1 protein expression, which was demonstrated to be three times higher for the M2 phenotype. This improvement in nanoparticles incorporation was not observed using those functionalized with the scrambled peptide. Moreover, it appears, from the literature [39], that there is no RNA or protein expression of NRP-1 in monocytes (THP-1).

A PDT strategy targeting M2 macrophages could appear very relevant due to their main localization in the peri necrotic and proliferating zones [40]. Different studies assessed the influence of macrophage polarization on silica-based nanoparticle uptake [41,42]. Their conclusions are in contradiction to ours, as they did not use a targeting peptide that was over-expressed by the M2 phenotype [43]. Indeed, the M1 phenotype macrophage are usually tumor-resistant mainly due to their intrinsic phagocytosis, leading to nanoparticle uptake [41,44]. By focusing on control nanoparticle uptake, we demonstrated that M0 macrophages incorporated more AGuIX@PS@scramble, illustrating their phagocytic ability, as was already described [45]. When using, for instance, Temoporfin^®^ (cyclodextrin in liposome nanoparticles), the results were also similar, with higher uptake by the M0 macrophages compared to M2 [18]. In our case, AGuIX@PS@KDKPPR nanoparticle uptake by M2 macrophages expressing NRP-1 protein seems to be a very relevant result for the selective destruction of these protumor macrophage populations. Moreover, we also evidenced that post-PDT, the U87 secretome polarized the macrophages toward an M1 phenotype. This polarization towards a proinflammatory phenotype seems relevant as it would allow for sustained tumor growth control after PDT. Indeed, a culture of post-PDT tumor cells (B16F10, murine melanoma) with nonactivated or activated macrophages (RAW) in M1 or M2 phenotypes led to different effects on tumor growth [46]. The authors demonstrated that when cells were incubated with M0 or M2 macrophages, tumor cell viabilities increased by 1.9 and 2.6 times, respectively, compared to the controls (tumor cells alone), highlighting the protumor properties of TAM. On the contrary, the viability of tumor cells decreased by 26% when cultured with the M1 macrophages, indicating the antitumor efficacy of M1-like macrophages.

As already described by other teams, we observed in U87 glioma xenografts in vivo, a protumoral environment characterized by a majority of M2 phenotype that could promote angiogenesis and tumor malignancy [47,48]. Interestingly, when using AGuIX@PS@KDKPPR nanoparticles in vivo, we previously visualized tumor tissue by T1-weighted imaging and illuminated it by interstitial stereotactic PDT [5]. When compared to the control group (using AGuIX@PS nanoparticles without peptide), we highlighted that the percentage of tumors not having reached two times their initial volume had increased, leading to a statistically significant increase in prolonged tumor growth delay of 13 days with KDKPPR peptide instead of 7 days without functionalization using this affin peptide [5]. As ADC values can reflect acute inflammatory reactions and inflammatory severity, by using the same in vivo PDT protocol as was previously published [5], in this study, we demonstrated that the increase in ADC values post-treatment was related to an inflammatory response (cytotoxic and vasogenic edema). It could also be linked to cellular necrosis [49,50]. In preclinical studies, post-PDT inflammatory changes influence the antitumor adaptive immune response [51,52,53]. We also confirmed an acute inflammatory response post-PDT, labeling CD-68 macrophages; the invasion of myeloid cells by tumors was already reported in other preclinical and clinical studies [54,55]. Moreover, Vidyarthi et al. demonstrated (using Kaplan–Meir survival plots) that patients with low M2 marker expression exhibited better survival rates, indicating that M2 targeting is an effective and promising strategy [56].

## 5. Conclusions

PDT using AGuIX@PS@KDKPPR nanoparticles promotes newly recruited macrophage polarization toward the M1 phenotype, which displays proinflammatory and antitumor properties. Moreover, by targeting M2 macrophages, we can eliminate M2 pro-tumor macrophages. As interstitial PDT applied to GBM is already suggested in clinical practice, our current goal is to revise the existing practice by exploiting NRP-1-targeting nanoparticles, i.e., to propose an optimized irradiation scheme by taking into account inflammation and vascular responses. We might suggest an irradiation protocol, leading first to the initial destruction of M2 macrophages, followed by a fractionated iPDT to capitalize on the polarization in M1 post-treatment (Figure 10).

The pre-PDT phase: the majority of macrophages are of the M2 phenotype (in red), leading to tumor proliferation.

Early post-PDT phase: thanks to M2 targeting (high NRP-1 expression), the remaining macrophages will be of the M1 phenotype (in yellow), leading to tumor destruction.

Late post-PDT phase: PDT effect on macrophage polarization is time-dependent; an increase in M2 macrophages will be observed [54]. These successive phases argue for fractionated PDT.

NRP-1 functions as a receptor on M2 macrophages, and a targeting strategy using AGuIX-KDKPPR nanoparticles could also be extended to other approaches, notably diagnostics, perfecting tumor detection, and monitoring therapeutic responses.

## Figures and Tables

**Figure 1 pharmaceutics-15-00997-f001:**
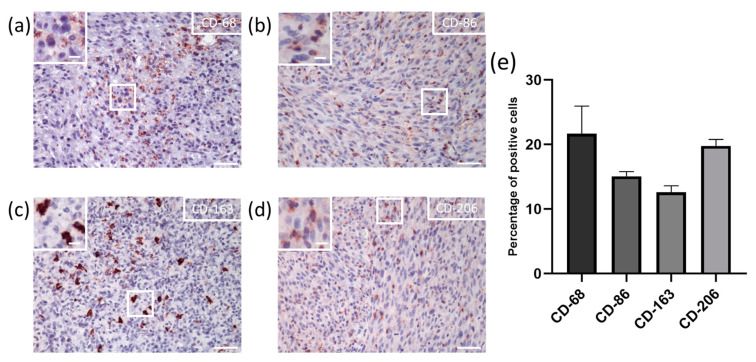
In vivo macrophage population in U87 tumor tissue. General view of (**a**) CD-68, (**b**) CD-86, (**c**) CD-163, and (**d**) CD-206 immuno-histochemical staining (scale = 50 µm in major squares; 10 µm in minor squares); (**e**) percentage of M2 macrophages infiltration throughout the tumor tissue. Mean percentage of positive cells (*n* = 3, mean ± SD).

**Figure 2 pharmaceutics-15-00997-f002:**
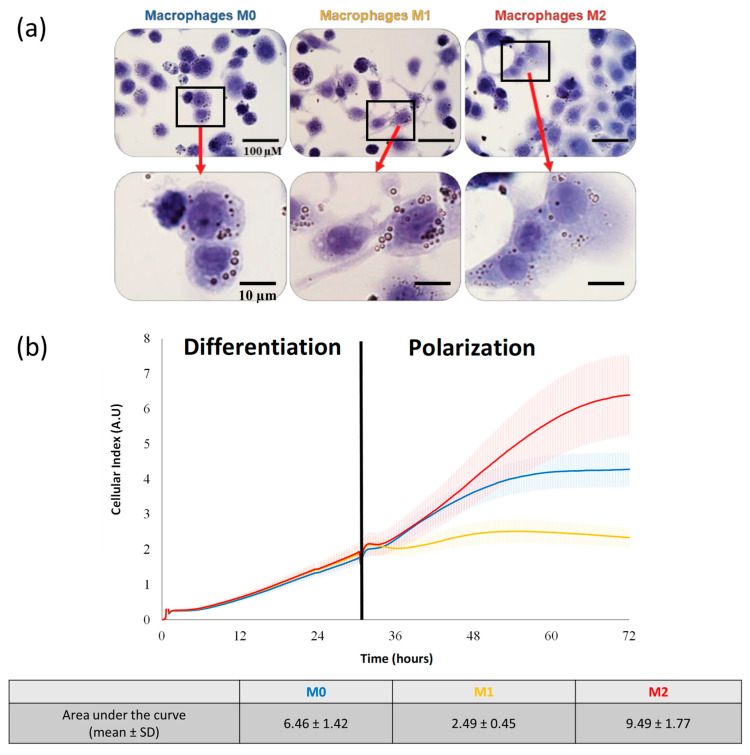
Morphological appearance and adhesion capacity of macrophages. (**a**) Morphological changes by optical image microscopy were assessed seven days after the polarization process. (**b**) Adhesion capacities by impedance measurements. The mean values of the cell index are represented by a solid line, integrating the variability of the measurements (6 replicates/group). The area under the curve values (mean ± SD) were calculated between 24 h and 72 h during the polarization phase.

**Figure 3 pharmaceutics-15-00997-f003:**
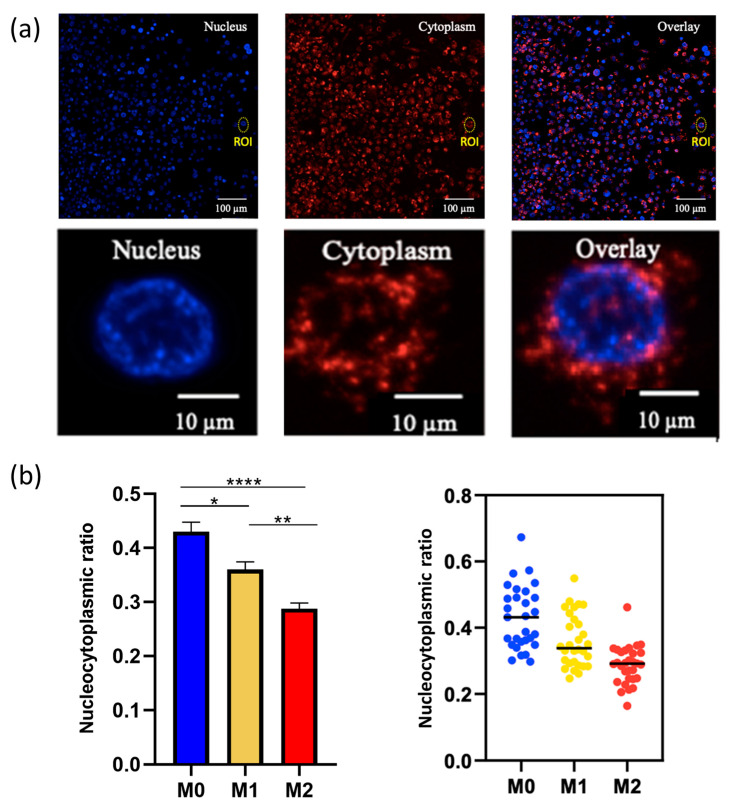
Macrophages nucleocytoplasmic ratio. (**a**) Using ImageJ software and selecting regions of interest (ROI), the nucleocytoplasmic ratio values (nucleus/cytoplasm volumes) were calculated for 30 cells/macrophage phenotype. (**b**) Macrophage nucleocytoplasmic ratio values (M0 in blue, M1 in yellow, and M2 in red). Mean ± SD and statistical analysis (Kruskal–Wallis, * *p* < 0.01; ** *p* < 0.001; **** *p* < 0.0001).

**Figure 4 pharmaceutics-15-00997-f004:**
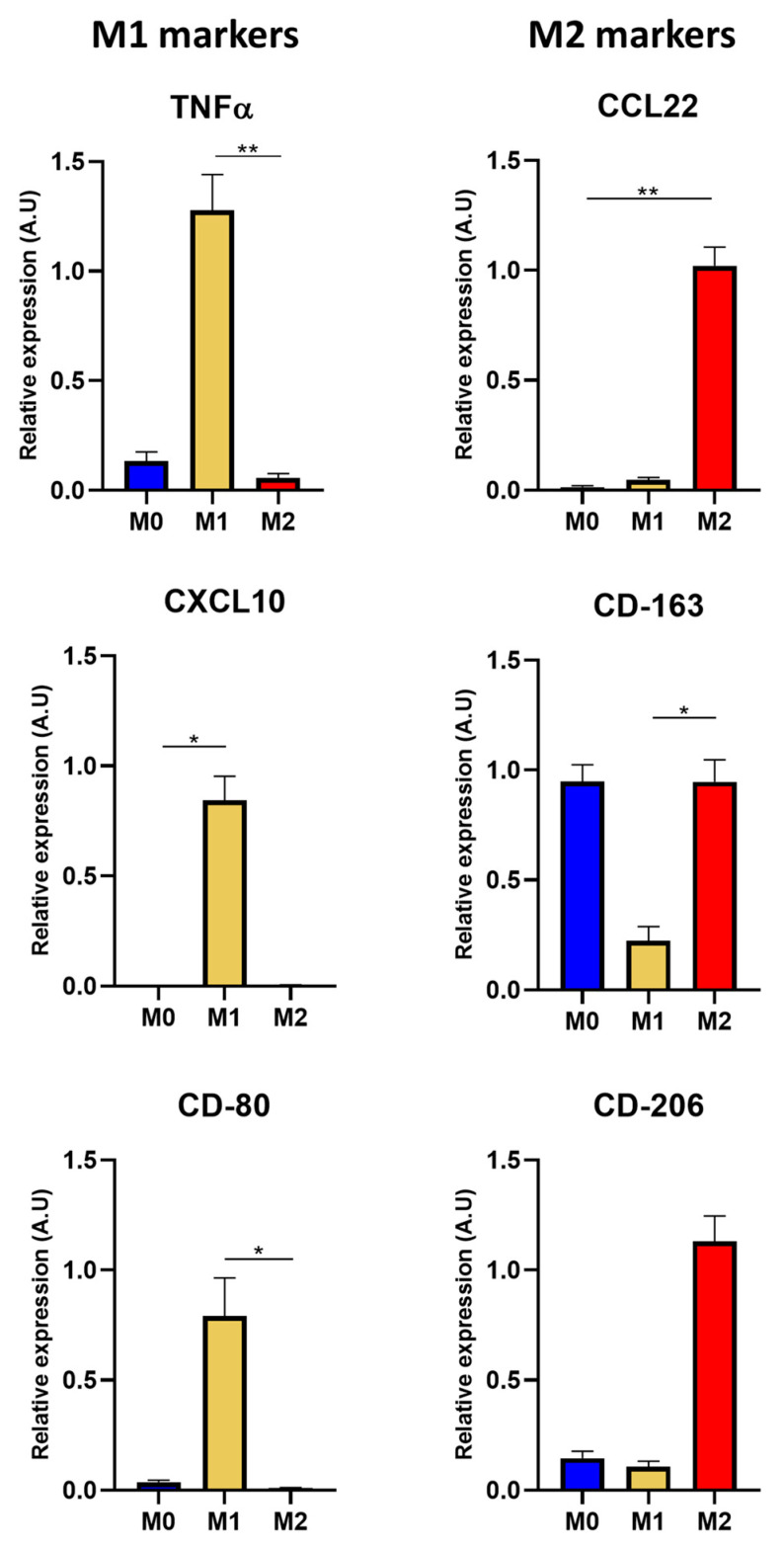
Gene expression of macrophage phenotypes. ARN extraction was performed 24 h after THP-1 polarization protocol. Mean relative gene expression of markers for M1 and M2 (*n* = 3 for each protein marker, mean + SD, * *p* < 0.01; ** *p* < 0.001 with Kruskal–Wallis test).

**Figure 5 pharmaceutics-15-00997-f005:**
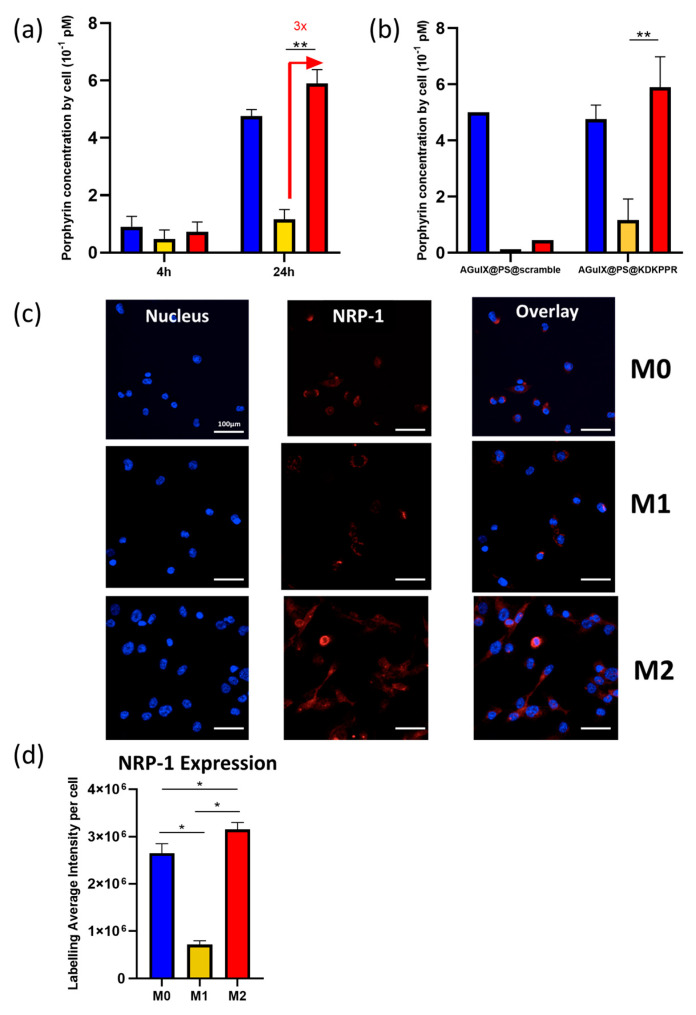
Nanoparticles uptake regarding NRP-1 expression in macrophage populations. Spectrofluorometric measurement of (**a**) AGuIX@PS@KDKPPR uptake (1μM) by macrophages (M0, M1, and M2) as a function of time (4 h or 24 h) and (**b**) AGuIX@PS@KDKPPR and AGuIX@PS@scramble 24 h post-incubation (*n* = 5, mean + SD). Statistics were carried out using Kruskal–Wallis test (** *p* < 0.001). (**c**) NRP-1 protein expression in macrophages was assessed by immune-cytofluorescence. NRP-1 protein expression is visible in red, and nuclei were labeled with Hoechst (in blue). (**d**) Quantification using MetaXpress software was carried out on ICF images, measuring the average labeling intensity per cell (101 cells for M0, 71 cells for M1, and 288 cells for M2). Mean + SD, * *p* < 0.0001.

**Figure 6 pharmaceutics-15-00997-f006:**
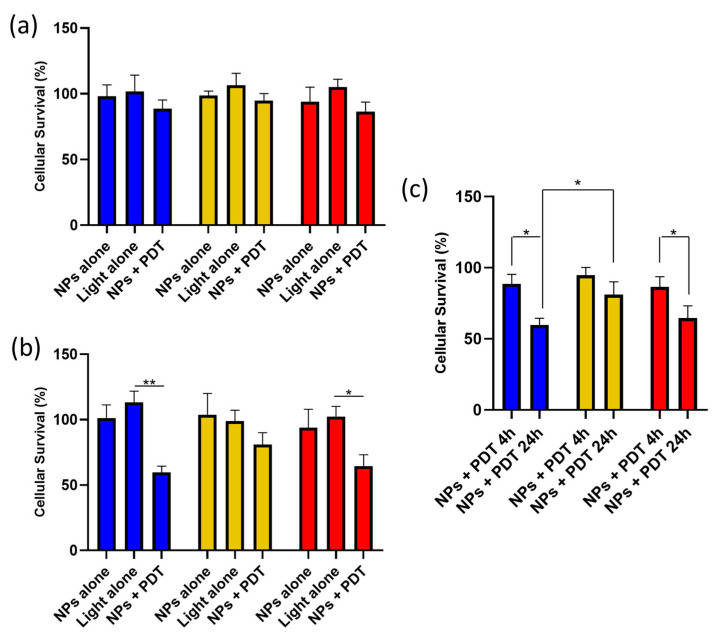
Cell survival on macrophages post-PDT. AGuIX@PS@KDKPPR nanoparticles (NPs, 1µM) were incubated with M0, M1, and M2 macrophages for (**a**) 4 h or (**b**) 24 h (group “NPs + PDT”). Several controls were performed, including “NPs alone” and “Light alone”. Cell survival was assessed 24 h post-PDT (10 J·cm^−2^, 4.54 mW·cm^−2^). Macrophage phenotype (M0 in blue; M1 in yellow; M2 in red) in columns (*n* = 4, mean + SD; * *p* < 0.01; ** *p* < 0.001, by Kruskal–Wallis test).

**Figure 7 pharmaceutics-15-00997-f007:**
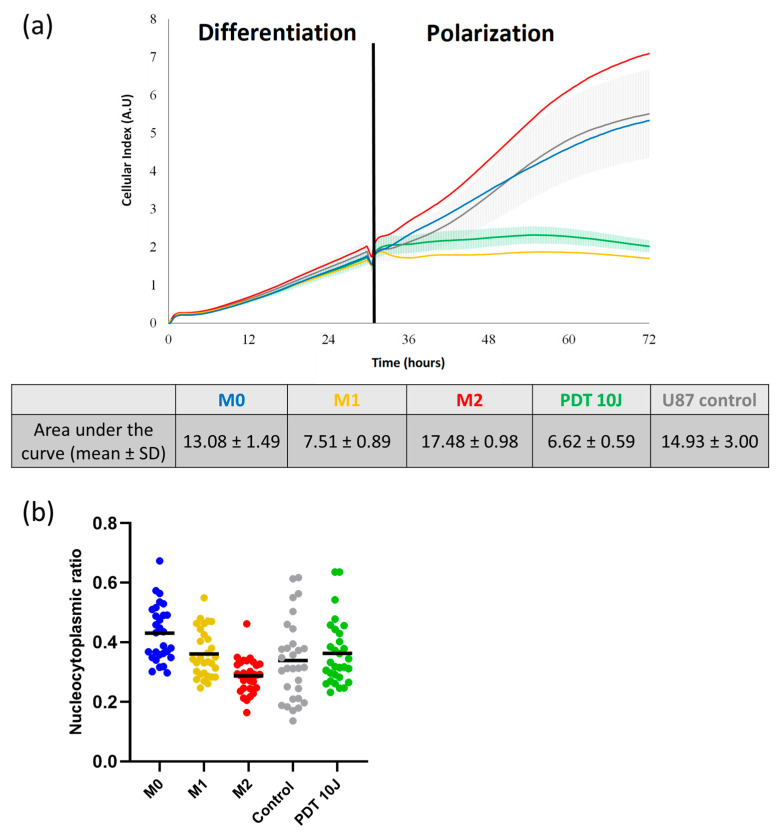
Influence of U87 secretome pre- and post-PDT on macrophage polarization. Impedance measurements were performed for 72 h, including the differentiation (24 h) and polarization (48 h) phases. (**a**) After the differentiation step, macrophages were contacted with secretome from U87 cells or from U87 post-PDT using AGuIX@PS@KDKPPR (1 µM) at a fluence of 10 J·cm^−2^. The mean is shown as a solid line with the variability of the replicate measurements (4 replicates/group). Area under the curve was calculated between 24 h and 72 h during the polarization phase. (AU arbitrary unit). (**b**) Macrophages nucleocytoplasmic ratio values for M0, M1, and M2 macrophages compared to macrophages in contact with secretome of U87 cells pre-PDT (Control) and post-PDT (PDT 10J) (30 cells/condition). Each point represents one cell (*n* = 30).

**Figure 8 pharmaceutics-15-00997-f008:**
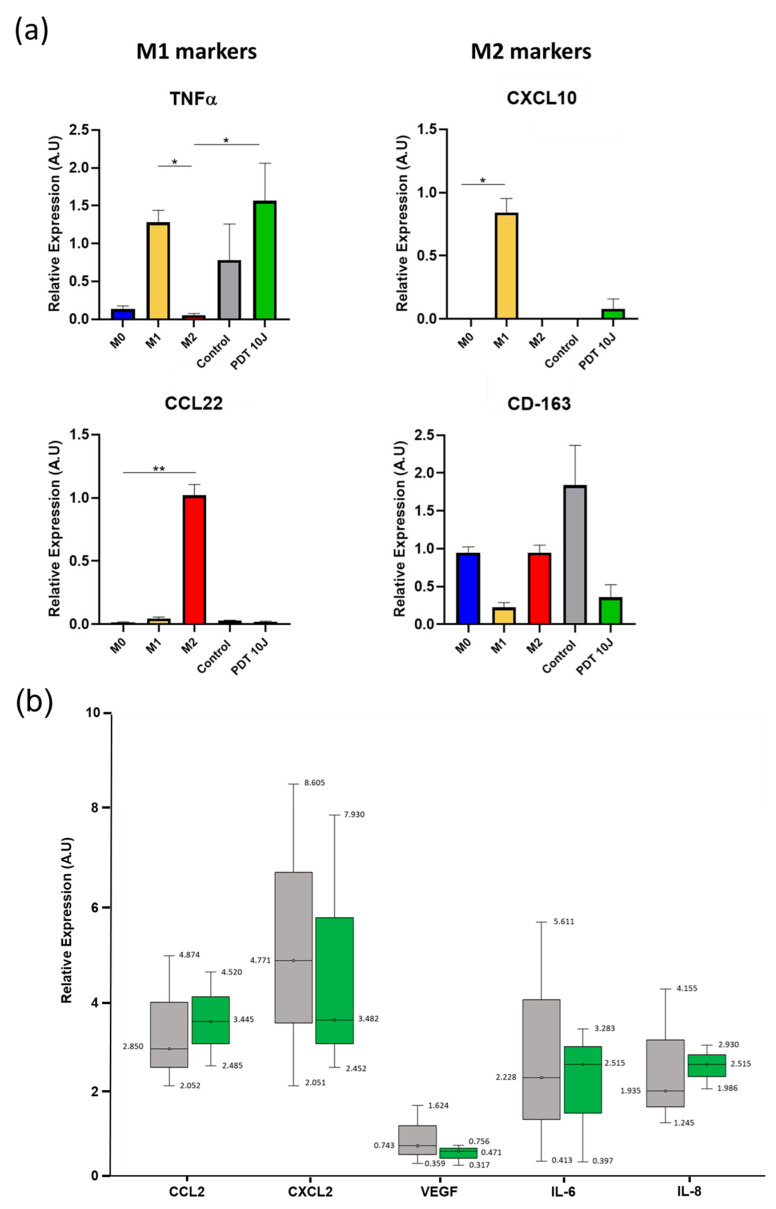
Effect of U87 secretome pre- and post-PDT on macrophage gene expression. (**a**) ARN extractions were performed 24 h after THP-1 polarization protocol. RT-qPCR was carried out on samples at 25 ng·μL^−1^ using SYBR Green as a fluorescent reporter for quantification. The mean relative gene expression of the markers for M1 and M2 (*n* = 3, mean + SD, * *p* < 0.01; ** *p* < 0.001 by Kruskal–Wallis test). (**b**) Secretome analysis by Proteome Array, pre- (in grey) and post-PDT (in green) (*n* = 3). We analyzed several cytokines, including CCL2, CXCL2 (macrophage recruitment), VEGF, IL-6, and IL-8 (pro-tumoral).

**Figure 9 pharmaceutics-15-00997-f009:**
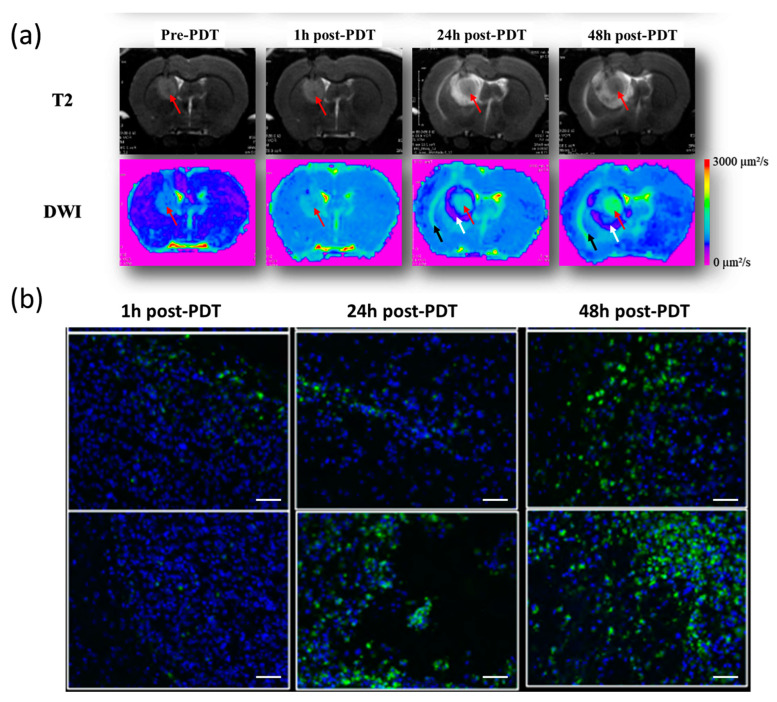
In vivo post-PDT inflammation. (**a**) Rats treated by PDT after intravenous injection of AGuIX@PS@KDKPPR nanoparticles (1.75 μmol·kg^−1^ porphyrin equivalent) and followed by MRI analysis via T2 anatomical and diffusion (DWI—diffusion-weighted imaging) sequences. Tumor indicated by a red arrow. On DWI, black and white arrows indicate the presence of vasogenic and cytotoxic edema, respectively. (**b**) Macrophage infiltration, demonstrated by immunofluorescence. Cell nuclei labeled with Hoechst (blue) and macrophages with CD-68 antibodies (green), ×20 (scale bars = 50 µm).

**Figure 10 pharmaceutics-15-00997-f010:**
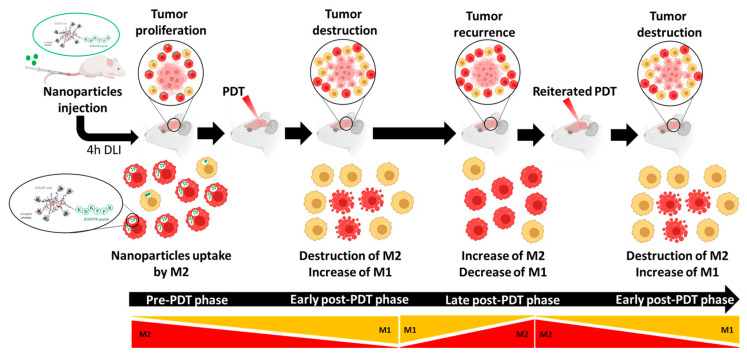
Workflow proposal for PDT integrating macrophage polarization and NRP-1 targeting. Four hours (drug light interval (DLI)) after intravenous injection of AGuIX@PS@KDKPPR nanoparticles, interstitial PDT is performed.

**Table 1 pharmaceutics-15-00997-t001:** qPCR primers designed on Primer-BLAST and verified on BLAST-N (NCBI).

Marker	Gene	Forward 5′-3′	Reverse 3′-5′	Length (bp)	NM
M1	TNFα	CCTCAGCCTCTTCTCCTTCC	GGTTGCTACAACATGGCT	191	NM_000594.4
	CXCL10	GAGTCTCAGCACCATGAATCAA	CAGTTCTAGAGAGAGGTACTCCTTG	95	NM_001565.4
	CD-80	CATCTGACGAGGGCACATAC	GGTGTAGGGAAGTCAGCTTTG	112	NM_005191.4
M2	CCL22	GCGTGGTGTTGCTAACCTTC	CCACGGTCATCAGAGTAGGC	115	NM_002990.5
	CD-163	TCACAATGAAGATGCTGGCG	CCTGCAAACCACATCAGCTT	169	NM_004244
	CD-206	ACCTCACAAGTATCCACACCATC	CTTTCATCACCACACAATCCTC	213	NM_002438.4
Reference	GAPDH	AGTCAGCCGCATCTTCTTTT	CCAATACGACCAAATCCGTTG	97	NM_002046.7

**Table 2 pharmaceutics-15-00997-t002:** Cytokines detected by the Human Cytokine Antibody Array kit.

C-X-C Motif Chemokines	C-C Chemokines	Interleukins	Growth Factors	Other Cytokines
CXCL1; CXCL2; CXCL5; CXCL8; CXCL9; CXCL12	CCL1; CCL2; CCL5; CCL7; CCL8; CCL15; CCL17; CCL22	IL-1α; IL-6; IL-1β; IL-7; IL-2; IL-10; IL-3; IL-12; IL-4; IL-14; IL-5; IL-15	PDGF BB; TGF-β; VEGF; IGF-1; EGF	IFNγ; TNFα; TNFβ; G-CSF; M-CSF; GM-CSF; Angiogenin; Thrombopoietin; Leptin; Oncostatin M; SCF

CXCL: chemokine C-X-C motif Ligand; CCL: C-C motif chemokine ligand; IL: Interleukin; PDGF: Platelet Derived Growth Factor; TGF: Transforming Growth Factor; VEGF: Vascular Epithelial Growth Factor; IGF: Insulin-like Growth Factor; EGF: Epithelial Growth Factor; IFN: Interferon; TNF: Tumor Necrosis Factor; CSF: Colony Stimulating Factor; SCF: Stem Cell Factor.

## Data Availability

Data available on request. The data presented in this study are available on request from the corresponding author.

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
