# Peer review of "Targeting Glioblastoma-Associated Macrophages for Photodynamic Therapy Using AGuIX®-Design Nanoparticles"

_pharmaceutics, 2023, doi:10.3390/pharmaceutics15030997_

Round 1

Reviewer 1 Report

The article by Lerouge and coauthors claims as main aim “to characterize in vitro the impact of macrophages NRP-1 protein expression on the uptake of functionalized AGuIX®-design nanoparticles and to describe the influence of GBM cells secretome post-PDT on macrophages polarization into M1 or M2 phenotype” (Lines 24-27 of the Abstract). Unfortunately, according to my opinion, the experiments presented fail in accomplishing these goals. The experiments demonstrate that the nanoparticles with the targeting peptide are incorporated better in M0 and M2 macrophages, than in M1 macrophages, but using the scrambled peptide, results are not so obviously indicating involvement of NRP-1.  Moreover, not all the experiments with U87 secretome report a comparison between untreated and PDT-treated U87, to the detriment of the results themselves.

There is also a disequilibrium (starting from the Abstract) between the first part of the article on the characterization of the macrophage population that is mostly preliminary to the subsequent experiments, and the second one focusing on NRP-1 expression and nanoparticle uptake. Accordingly, the Introduction seems to me a bit too long and confusing, not focusing on PDT methodology and effect together with NRP-1 expression. The Discussion is sometimes redundant with the Introduction. Finally, the description of the experiments and of the results obtained is often very concise and should be improved.

I would also recommend English language editing and proofreading.

Below are the specific comments:

1)    The title of paragraphs 3.1 and 3.2 are too general. The experimental context is not clearly introduced (see line 326).

Figure 1. I would suggest increasing the brightness of the images so that signals will be more evident. Please insert magnification bars also in the squares.

Please indicate what CD68 marks. (lines 330-331)

To analyze macrophage distribution, markers for M1 and M0 macrophages should also be used. Moreover, how is it possible that the percentage of positive cells for a marker for a subtype of M2 macrophage (CD-86) is higher than a marker for M2 macrophages, such as CD-163? What does it mean n=3? (line 336) Is the number of samples or the number of sections? How many images for each section were analyzed?

2)    Figure 2a: please indicate the magnification bar also in the lower images at higher magnification and use a square to clearly indicate the corresponding areas in the images at lower magnification.

Lines 364-366- Please clarify why these results are expected and the link to previous ones.

1)    Regarding the experiment of nanoparticle uptake, at 24h both M0 and M2 macrophages incorporated more particles with the targeting peptide, whereas using the scrambled peptide, M0 incorporation still stays high, whereas M2 decreases (figure 5a and b).

Since both M0 and M2 express high levels of NRP-1 (figure 5 c and d), approximately 3 times higher than M1 macrophages, how could authors explain this? I would expect the same behavior for both types of macrophages. I would suggest another type of experiment such as RNA interference to demonstrate NRP-1 involvement.

2)    Effect of PDT on macrophages

In Figures 6 a and b, PDT alone does not seem to have any effect on macrophage cell survival, nor at 4h nor at 24h. Is it expected? Or an even a small effect should be present?

In this and previous experiments, M0 and M2 macrophages behave in the same way using the targeting peptide, but in the previous one, the behavior with scrambled peptide is different. It would be really interesting to repeat also PDT experiment using the scrambled peptide and compare M0 and M2 macrophages.

The term overexpression (lines 421-422) is misleading. I would simply speak of high(er) expression in some macrophage (M0 and M2) subtypes with respect to others (M1). What about NRP-1 expression in monocytes? Moreover, I believe that is better to use “NRP-1 gene expression” or “NRP-1 protein detection” instead of “protein expression”.

3)    Please be aware that the U87 “secretome” or “conditioned medium” contains not only soluble factors but also, extracellular vesicles, biological nanoparticles that are usually released by tumor cells in general and for sure by U87 cells.

Which is the effect of PDT on U87 cells? Cell death for sure might influence the secretome effect and composition.

In the experiments in Figures 7b and 8, it would be extremely valuable to add also the sample relative to macrophages treated with no-PDT U87 secretome as control, otherwise, it is not possible to assess the effect of PDT. Moreover, since the two M1 markers used (TNFalfa and CXCL10) have contrasting behavior, it would be valuable to analyze also the expression of other M1 markers, such as CD-80 previously analyzed, and even M0 markers.

Lines 468-469: this sentence might be misleading, cause the two phenotypes refer to different types of experiments.

4)    In the in vivo experiment I would suggest analyzing by IHC the subtype (M1 or M2) of the infiltrated macrophage to support in vitro data.

5)    The beginning of the Discussion (lines 490-510) and Introduction (lines 40-68) are very similar. Redundancy is also evident in other parts. Please revise. The final part of the Introduction (lines 105-117) could be also moved to the Discussion.

In the Discussion, the authors report results of a proteome array for secretome analysis as data not shown (lines 544-546), not previously mentioned in the Results. The results might be relevant, but they have to be inserted in The Results (and Methods) with relative Figures, otherwise, they should be deleted.

Sometimes there is a bit of confusion between the discussion of data in the present article and  of previous articles by the same authors (i.e. lines 580-599 at the end of the discussion)

Please revise the Conclusions according to all the comments. 

Reviewer 2 Report

Overall, while this study provides valuable insights into the role of macrophages and NRP-1 protein expression in GBM and the potential of PDT and multifunctional nanoparticles for GBM treatment, further research is needed to validate the findings and address the limitations of this study.

The authors need to provide clarification on the limitations of this work and how they would address this in future work:

There are several potential shortcomings or limitations of this study that should be considered:

1.      In vitro study: This study was performed in vitro using THP-1 human monocytes, which may not fully reflect the complex in vivo environment and interactions in human GBM. The results of this study should be interpreted with caution and further studies are needed to confirm the findings in vivo.

2.      Small sample size: The sample size used in this study may not be sufficient to draw definitive conclusions, as the findings may be influenced by individual variations or experimental errors. Larger studies with more samples are needed to validate the results.

3.      Lack of clinical data: Although this study highlights the potential of PDT and multifunctional nanoparticles for the treatment of GBM, the findings are based on in vitro experiments. The efficacy and safety of this approach in clinical practice need to be further evaluated in clinical trials.

4.      Limited scope: This study focused on the impact of macrophage NRP-1 protein expression on the uptake of functionalized AGuIX®-design nanoparticles and the influence of GBM cells secretome post-PDT on macrophage polarization into M1 or M2 phenotype. The study did not address other factors that may affect the effectiveness of PDT and multifunctional nanoparticles in treating GBM, such as the tumor microenvironment, drug resistance, and potential adverse effects.

Minor comments:

1.      Figure 9 (b). Need to provide magnification and scale bars

Reviewer 3 Report

The work presented by Lerouge et al is very interesting and promising in the field of GBM treatment. Experiments using THP-1 are interesting but the in vivo PDT experiments with GMB rats are promising. Concerning these in vivo experiments Is n=3 enough for the group of rats exposed to light after injection of AGuIX@PS@KDKPPR? The other 2 groups of rats are n=5 and n=6 that a common size in animals to obtain statistical significance data.

And after PDT experiments in figure 9 you are showing images of only one animal, could you add as a supplementary file the images of the other 2 animals? And, Can you quantify the levels of CD-68 in the three animals?

And information about the inflammation and CD-68 levels of animals treated with NPs without PDT and with NPs with the random peptide and PDT will be very valuable to highlight the efficacy of NPs with NPR-1 targeting and PDT.

Round 2

Reviewer 1 Report

The authors greatly improved the article and answered most of my comments. Below are some points still open:

1)    Please specify on line 383 “Tumor tissue of U87 xenograft rat models”

2)  Please report also in the manuscript the following explanation on CD68 marker

“CD-86 to be both M1 and M2b markers (Shapouri-Moghaddam et al., J. of Cellular Physiology, 2018; Wang et al., J. of Leukocyte Biology, 2019), which could explain its higher expression level compared to the one found for CD-163”.

3)    Please insert in the manuscript this detail on NRP-1 expression. “Regarding NRP-1 expression (transcripts or protein) in monocytes (THP-1), no experiment was performed, but it appears from the literature that there is no RNA or protein expression of NRP-1 in monocytes (Kawaguchi et al., Cancer immunology immunotherapy, 2017).

4)    Please after the experiment on secretome, at line 507 not speak only of soluble factors, as you agree in response n. 5.

Author Response

All your requirements have been taken into account in the manuscript.

With warm thanks.

Reviewer 2 Report

I am commend the authors for the excellent revision and addressing all my comments. I have no further comments.

Author Response

With warm thanks

Reviewer 3 Report

Congratulations

Author Response

With warm thanks